# Interannual variations of water vapor in the tropical upper troposphere and the lower and middle stratosphere and their connections to ENSO and QBO

Edward W Tian[1,2], Hui Su[1], Baijun Tian[1], and Jonathan H. Jiang[1]

[1]Jet Propulsion Laboratory, California Institute of Technology, Pasadena, CA, USA
[2]Department of Economics, University of California, Santa Cruz, CA, USA

*Correspondence to*: Dr. Jonathan Jiang (Jonathan.H.Jiang@jpl.nasa.gov)

**Abstract.** In this study, we analyze the Aura Microwave Limb Sounder water vapor data in the tropical upper troposphere and the lower and middle stratosphere (UTLMS) (from 215 hPa to 6 hPa) for the period from August 2004 to September 2017 using time-lag regression analysis and composite analysis to explore the interannual variations of tropical UTLMS water vapor and their connections to El Nino Southern Oscillation (ENSO) and quasi-biennial oscillation (QBO). Our analysis shows that the interannual tropical UTLMS water vapor anomalies are strongly related to ENSO and QBO which together can explain more than half (~50-60%) but not all variance of the interannual tropical water vapor anomalies. We find that ENSO's impact is strong in the upper troposphere (~215 to ~120 hPa) and near the tropopause (~110 to ~90 hPa) with a ~3-month lag but weak in the lower and middle stratosphere (~80 to ~6 hPa). In contrast, QBO's role is large in the lower and middle stratosphere with an upward propagating signal starting at the tropopause (100 hPa) with a ~2-month lag, peaking in the middle stratosphere near 15 hPa with a ~21-month lag. The phase lag is based on the 50-hPa QBO index used by many previous studies. This observational evidence supports that the QBO's impact on the tropical stratospheric water vapor is from its modulation on the tropical tropopause temperature and then transported upward with the tape recorder as suggested by many previous studies. In the upper troposphere, ENSO is more important than QBO for the

interannual tropical water vapor anomalies that are positive during the warm ENSO phases but negative during the cold ENSO phases. Near the tropopause, both ENSO and QBO are important for the interannual tropical water vapor anomalies. Warm ENSO phase and westerly QBO phase tend to cause positive water vapor anomalies while cold ENSO phase and easterly QBO phase tend to cause negative water vapor anomalies. As a result, the interannual tropical water vapor anomalies near the tropopause are different depending on different ENSO and QBO phase combinations. In the lower and middle stratosphere, QBO is more important than ENSO for the interannual tropical water vapor anomalies. For the westerly QBO phases, interannual tropical water vapor anomalies are positive near the tropopause and in the lower stratosphere but negative in the middle stratosphere and positive again above. Vice versa for the easterly QBO phases.

## 1 Introduction

Water vapor (WV) is the dominant greenhouse gas in the atmosphere and plays an important role in global weather and climate systems. Since higher temperature is associated with higher saturation vapor pressure, water vapor has a positive feedback to surface warming. Previous studies indicate that the water vapor feedback is the largest positive feedback in climate models that increases the sensitivity of surface temperature to increasing carbon dioxide (Held and Soden, 2000; Soden and Held, 2006). The middle and upper tropospheric water vapor dominates the water vapor feedback (e.g., Held and Soden, 2000; Riese et al., 2012; Soden et al., 2008). The stratospheric water vapor may account for about 10% of total water vapor feedback (Dessler et al., 2013). In addition, the water vapor in the stratosphere plays an important role in stratospheric ozone chemistry and global radiative balance (Forster and Shine, 1999; Solomon et al., 2010). Thus, it is important to study the water vapor variability in the upper troposphere and the lower and middle stratosphere (UTLMS).

The water vapor in the tropical upper troposphere is mainly from the tropical lower troposphere through convective transport of water vapor and evaporation of convectively-transported or in-situ-produced cloud ices (e.g., Su et al., 2006; Tian et al., 2004). The majority of the water vapor in the tropical lower and middle stratosphere (LMS) is from the tropical upper troposphere through the tropical tropopause. The tropical LMS water vapor concentration is thus mainly determined by the tropical tropopause (~100 hPa) temperature that regulates the amount of tropical upper tropospheric water vapor entering the tropical stratosphere (e.g., Fueglistaler et al., 2009; Liang et al., 2011; Randel et al., 2004; Zhou et al., 2001, 2004). Part of the water vapor in the tropical LMS is also from the local methane oxidation.

The water vapor in the tropical UTLMS exhibits multi-timescale variations ranging from daily to decadal (e.g., Fueglistaler et al., 2009; Fujiwara et al., 2010; Hegglin et al., 2014; Jiang et al., 2015; Mote et al., 1996; Schwartz et al., 2008; Solomon et al., 2010; Tian et al., 2004; 2006; 2010). In particular, large interannual variations of water vapor in the tropical UTLMS have been observed and shown to be important for both climate and chemical reasons (e.g., Dessler et al., 2013; 2014; Fueglistaler and Haynes, 2005; Liang et al., 2011; Liess and Geller, 2012; Randel and Jensen, 2013; Tao et al., 2015; Ye et al., 2018). Several well-known interannual climate variabilities and their interactions are found to modulate the interannual variations of tropical UTLMS water vapor, such as the El Nino Southern Oscillation (ENSO) (e.g., Dessler et al., 2014; Liang et al., 2011; Randel et al., 2004; Su and Jiang, 2013; Ye et al., 2018), the quasi-biennial oscillation (QBO) (e.g., Dessler et al., 2014; Fueglistaler and Haynes, 2005; Geller et al., 2002; Kawatani et al., 2014; Liang et al., 2011; Liess and Geller, 2012; Randel et al., 2004; Randel and Jensen, 2013; Tao et al., 2015; Ye et al., 2018), and the interannual variations in the strength of the Brewer–Dobson circulation (BDC) (e.g., Dessler et al., 2013; 2014; Randel et al., 2006; Ye et al., 2018).

The BDC is a slow stratospheric mean meridional circulation in which air parcels rise in the tropics, drift poleward into the stratosphere, and are transported downward in the high-latitude regions via its shallow and deep branches (Brewer, 1949; Butchart, 2014). The BDC is one of the few truly global-scale phenomena observed in the Earth's atmosphere below ~50 km. It is particularly prominent because of its widespread controlling influence on the stratosphere. For instance, it has important roles in determining the thermodynamic balance of the stratosphere, the temperature of the tropical tropopause, the water vapor entry into the stratosphere, the period of the tropical QBO, the lifetimes of CFCs and some greenhouse gases, and the transport and redistribution within the stratosphere of greenhouse gases, ozone, aerosols, volcanic and radioactive debris (Butchart, 2014). Driven by wave breaking in the stratosphere, the BDC varies on subseasonal to decadal timescales.

The QBO is a major mode of interannual variability in the tropical upwelling of the BDC (Baldwin et al., 2001; Lindzen and Holton, 1968). The QBO describes the quasi-biennial oscillation of downward propagating easterly or westerly zonal winds in the equatorial stratosphere from the middle stratosphere to the tropopause with a period of ~28 months (Baldwin et al., 2001). It is well known that the QBO causes large-scale circulation changes that affect ozone, water vapor, methane and global weather and climate. The QBO's impact on the tropical UTLMS water vapor is mainly through the QBO's influence on the tropical tropopause temperature that regulates the amount of upper tropospheric water vapor entering the stratosphere (e.g., Fueglistaler et al., 2009; Fujiwara et al., 2010; Geller et al., 2002; Kawatani et al., 2014; Liang et al., 2011; Randel et al., 2004; Zhou et al., 2001, 2004). Mostly driven by equatorially trapped waves, the QBO triggers an anomalous meridional circulation in the stratosphere between the tropics and subtropics (from the equator to 30N and 30S) to maintain the thermal wind balance between the descending QBO wind shear and its

temperature anomaly (Diallo et al., 2018; Tweedy et al., 2017). At the equator, westerly shear (westerlies aloft and easterlies below) is in balance with a downward-propagating and adiabatically warmed perturbation, while easterly shear (easterlies aloft and westerlies below) produces an upward-propagating and adiabatically cooled perturbation. The tropical upwelling perturbation is anticorrelated with the tropical temperature perturbation in the lower stratosphere. The enhanced upwelling during easterly shear and reduced upwelling during westerly shear in the tropics are mass balanced by the changes in the subtropical descent. The circulation is "completed" by the equatorial divergence/convergence of air at the levels of maximum easterly/westerly winds (Choi et al., 2002). As the westerly shear reaches the tropopause, it warms the tropopause and increases the amount of the upper tropospheric water vapor entering the lower stratosphere. Conversely, as the easterly shear reaches the tropopause, it cools the tropopause and decreases the amount of the upper tropospheric water vapor entering the lower stratosphere (Diallo et al., 2018; Tweedy et al., 2017). It is also noted that the QBO modulates the extratropical wave activity, an important driver for the BDC, which influences the tropical cold point tropopause temperature.

ENSO is the interannual oscillation (2-7 year) of sea surface temperatures (SSTs) and easterly trade winds in the tropical Pacific ocean caused by the coupled interactions between the ocean and atmosphere (Wallace et al., 1998). It is the primary source of global interannual climate variabilities (Philander, 1990; Wallace et al., 1998). During a warm ENSO (El Niño) phase, trade winds are weaker and warm waters move eastward to the equatorial central and eastern Pacific. During a cold ENSO (La Niña) phase, trade winds are stronger and warm waters move further westward to the equatorial western Pacific. ENSO can modulate the tropical UTLMS water vapor through several physical and dynamical processes, such as convective transport of tropospheric water vapor, evaporation of cloud ice, and the perturbations of tropical tropopause (~100 hPa) temperature (e.g., Dessler et al., 2014; Gettelman et al., 2001; Liang et al., 2011;

Ye et al., 2018; Zhou et al., 2001, 2004). In the tropical upper troposphere, ENSO modulates the water vapor mainly through the convective transport of lower tropospheric water vapor and evaporation of cloud ice. In the stratosphere including the tropopause region, ENSO modulates the water vapor mainly through its influence on the tropical tropopause temperature that regulates the amount of water vapor entering the stratosphere. From a zonal mean perspective, El Niño events induce a tropospheric warming and a stratospheric cooling with a node near the tropopause, strengthen the tropical upwelling of the BDC, and decrease the tropical lower stratospheric ozone (Calvo et al., 2010; Randel et al., 2009). Lower stratospheric water vapor, however, is predominantly controlled by cold point temperatures over the tropical western Pacific (Avery et al., 2017; Diallo et al., 2018). El Niño events are associated with warmer cold point temperatures over this region, thereby causing increased lower stratospheric water vapor (e.g., Avery et al., 2017; Calvo et al., 2010; Konopka et al., 2016). In contrast, La Niña events induce an opposite effect.

Many previous studies have significantly improved our knowledge about the interannual variations of the tropical UTLMS water vapor. For example, Liang et al. (2011) studied the atmospheric water vapor and temperature variability in the tropical UTLMS using merged Aqua Atmospheric Infrared Sounder (AIRS) and Aura Microwave Limb Sounder (MLS) temperature and water vapor record (August 2004 to March 2010). They found that both ENSO and QBO impact the tropical tropopause water vapor and the water vapor anomalies near the tropical tropopause are strongly dependent on the alignment of ENSO and QBO phases. Dessler et al. (2013; 2014) performed a multi-linear regression of the tropical lower stratospheric (82-hPa) water vapor variability to the QBO, BDC and tropospheric temperature (which is correlated with ENSO). They found that the tropical lower stratospheric water vapor lags QBO by about 3 months and lags BDC by 1 month based on the 50-hPa QBO index. Ye et al. (2018) performed a two-dimensional multivariate linear regression of the tropical

tropopause water vapor interannual variability to the QBO, BDC and tropospheric temperature as a function of latitude and longitude based on satellite observations and model simulations. They found that the evaporation of convective ice from increased deep convection as the troposphere warms plays an important role in the tropopause water vapor variability in addition to changing tropopause temperature. Ding and Fu (2018) found that the tropical central Pacific SST warming contributes significantly to enhanced convection and thus sudden drop of the lower stratospheric (83-hPa) water vapor around 2000. They suggested that the tropical central Pacific SST is another important driver of the lower stratospheric water vapor variability on inter-decadal time scales.

During the boreal winter 2015–2016, a strong El Niño event (among the three strongest El Niño events on record) (Huang et al., 2016) was aligned with a westerly QBO phase. This westerly QBO phase was abruptly disrupted well before completion by an easterly phase in January 2016 (Newman et al., 2016; Osprey et al., 2016). The interplay between both circulation anomalies caused large changes in trace gas transport, the climate implications of which are currently a topic of debate. Avery et al. (2017) argued that the most recent El Niño event significantly moistened the lower stratosphere due to convective ice lofting, with the QBO having only a small contribution. In contrast, Tweedy et al. (2017) attributed the lower stratospheric water vapor changes from spring to autumn to the 2015–2016 QBO disruption. Diallo et al. (2018) showed that the alignment of a strong El Niño event with westerly QBO in early boreal winter of 2015–2016 substantially increased water vapor in the tropical lower stratosphere (positive anomalies of more than 20%). The sudden shift in the QBO from westerly to easterly wind shear significantly decreased global lower stratospheric water vapor from early spring to late autumn and reversed the lower stratosphere moistening to the lower stratosphere drying (negative anomalies of close to 20%). They emphasized that the control of the lower stratospheric water vapor anomalies strongly depends on the interactions between

ENSO and QBO phases. The interaction of El Niño and the westerly QBO phase leads to large positive lower stratospheric water vapor anomalies, while the interplay between La Niña and easterly QBO phase leads to negative water vapor anomalies. During weak and moderate ENSO events, the water vapor anomalies are mainly controlled by the QBO phase.

However, the aforementioned studies mainly focused on a few specific level of the UTLMS layer, either 82 hPa (Dessler et al., 2013; 2014; Ding and Fu, 2018) or 100 hPa (Ye et al., 2018) or based on limited data periods (Avery et al., 2017; Diallo et al., 2018; Liang et al., 2011; Tweedy et al., 2017). A comprehensive investigation of the interannual variations of the

tropical water vapor in the whole UTLMS layer with a much longer period and their relationships to ENSO and QBO is still lacking. In addition, the relative importance of ENSO and QBO on the tropical UTLMS water vapor interannual variabilities at different levels has not been well investigated in the previous studies.

This study seeks to investigate the interannual variations of water vapor in the tropical UTLMS layer and their relationships to ENSO and QBO using the Aura MLS UTLMS water vapor data. We are particularly interested in the relative roles of ENSO and QBO in the interannual variations of water vapor in the tropical UTLMS layer at different levels. This study distinguishes itself from previous studies in three following ways: (1) The current study

investigates the interannual variations of water vapor in the whole tropical UTLMS layer from 215 hPa to 6 hPa instead of a couple of layers in the previous ones. (2) The Aura MLS UTLMS water vapor data of much longer length (August 2004 to September 2017) are used in the current study than the previous ones. (3) The relative importance of ENSO and QBO on the tropical UTLMS water vapor interannual variabilities for the entire UTLMS layer and at

different phase lags are more completely investigated in the current study than the previous ones. (4) This study will present some new observational evidence to better understand the role

of ENSO and QBO in the tropical UTLMS water vapor interannual variations, especially regarding the role of QBO and its tape recorder effect. (5) This study will also present a composite view of the tropical UTLMS water vapor interannual variations based on different combinations of ENSO and QBO phases.

The rest of this paper is organized as follows. Section 2 describes the MLS water vapor data and the analysis methods. Section 3 presents the results followed by summary and conclusions in Section 4.

## 2 Data and Methods

We use Version 4.2 Level 2 daily Aura MLS water vapor volume mixing ratio product as described in Read et al. (2007) and Livesey (2015) from 215 hPa to 6 hPa over the period of August 2004 to September 2017. The MLS water vapor data were averaged to monthly means and gridded onto 2.5x2.5 horizontal spatial grids. The MLS Level 2 data have a vertical resolution of ~3 km and horizontal resolutions of ~7 km across track and ~200–300 km along

track. The useful altitude ranges are at pressure (p) ≤ 316 hPa but we only use the water vapor data above 215 hPa because of larger uncertainty below 215 hPa altitude. The measurement uncertainties (including biases) are 20% in the upper troposphere (p>100 hPa) and 10% near the tropopause (~100 hPa) and in the stratosphere (p ≤ 100 hPa) (Lambert et al., 2007; Read et al., 2007). These measurement uncertainties are retrieval uncertainties and estimated based on

(1) the average difference between the simulated retrieval and truth file; (2) the average difference between MLS measurements and the air borne measurements. These uncertainties should not affect our results because we are interested in the interannual anomalies instead of its means. In addition, Hegglin et al. (2013) show that MLS zonal monthly mean water vapor show very good to excellent agreement with the multi-instrument mean (MIM) in comparison

between thirteen instruments, throughout most of the atmosphere (including the UTLS) with mean deviations from the MIM between +2.5% and +5%, making these random errors irrelevant for the average monthly zonal mean water vapor anomalies used in this study (Diallo et al., 2018). The Aura MLS water vapor data have been used extensively in atmospheric

process analysis studies and climate model evaluations (e.g., Dessler et al., 2013; 2014; Flury et al., 2012; Jiang et al., 2012; Liang et al., 2011; Liu et al., 2018; Solomon et al., 2010; Su et al., 2006; Takahashi et al., 2016; Uma et al., 2014; Wu et al., 2012). The MLS water vapor data are freely available through the Aura MLS project website (https://mls.jpl.nasa.gov).

Since we are mainly interested in the tropical UTLMS, we first averaged the MLS monthly water vapor data between 15°S/N, and along the entire latitude band ($wv_{t,p}$). Then, the tropical mean seasonal cycle ($wv_{m,p}$, 12 months) was calculated as the averages of the tropical MLS monthly water vapor data at each calendar month over the whole MLS data record. Next, de-seasonalized monthly tropical water vapor anomalies were obtained by removing the tropical

mean seasonal cycle from the tropical monthly water vapor data ($wv'_{t,p} = wv_{t,p} - wv_{m,p}$). Then, the interannual (2-7 years) tropical water vapor anomalies (or short-handed as anomalies for simplicity) were isolated through the difference between the 12-month and 42-month running means of the de-seasonalized monthly tropical water vapor anomalies to remove the high-frequency (e.g., synoptic, seasonal, intraseasonal, and annual) and low-frequency (e.g., solar

cycle and decadal) variabilities. Last, the interannual monthly tropical water vapor anomalies were converted into percentage deviations through dividing the interannual monthly tropical water vapor anomalies by the long-term tropical mean ($wv_p$) at the respective pressure level. The resulting interannual monthly tropical water vapor anomalies in percentage deviations are used throughout the analysis.

Using the difference of running means of different widths as a band-pass filter is effective. A 12-month running mean will remove the high-frequency variabilities (<2 years) and keep the interannual variability (2-7 years) and the low-frequency variabilities (>7 years). A 42-month running mean will remove the high-frequency variabilities (<2 years) and the interannual variability (2-7 years) and keep the low-frequency variabilities (>7 years). As a result, the difference between the 12-month and 42-month running means will remove both the high-frequency (<2 years) and low-frequency (>7 years) variabilities and keep the interannual variability (2-7 years) only. This simple approach of band-pass filter has been used in the previous studies related to the Madden-Julian Oscillation (e.g., Tian et al., 2006; 2007; 2011).

To represent ENSO phases, we use a bimonthly multivariate ENSO index (MEI) downloaded from National Oceanic and Atmospheric Administration (NOAA) Earth System Research Laboratory (ESRL) website (https://www.esrl.noaa.gov/psd/enso/mei/). After spatially filtering the individual fields into clusters, the MEI is calculated as the first unrotated Principal Component (PC) of all six observed fields combined including sea-level pressure, zonal and meridional surface winds, sea surface temperature, surface air temperature, and total cloudiness over the tropical Pacific collected and published in International Comprehensive Ocean-Atmosphere Data Set (ICOADS) (Wolter and Timlin, 2011). Positive MEI values indicate warm ENSO (El Nino) phases while negative MEI values indicate cold ENSO (La Nina) phases. In order to keep the MEI comparable, all seasonal values are standardized with respect to each season and to the 1950-93 reference period. The MEI is computed separately for each of twelve sliding bi-monthly seasons (Dec/Jan, Jan/Feb, ..., Nov/Dec). We use the MEI value of month(i-1) and month(i) as if it were the value for month(i) only as advised by the NOAA MEI website.

For QBO, we use the standardized anomaly of monthly zonal mean zonal wind at the Equator and 50-hPa (u50, m s$^{-1}$) based on the National Centers for Environmental Prediction (NCEP) /

National Center for Atmospheric Research (NCAR) reanalysis (CDAS) downloaded from NOAA NCEP Climate Prediction Center (CPC) website (http://www.cpc.ncep.noaa.gov/data/indices/qbo.u50.index). Positive u50 values denote westerly QBO phases while negative u50 values denote easterly QBO phases. This 50-hPa

QBO index has been frequently used by previous studies (Dessler et al., 2013; 2014; Ye et al., 2018). The ENSO and QBO indices from August 2004 to September 2017 are shown in Fig. 1.

With the ENSO, QBO, and MLS data sets we conducted two types of analysis: lead-lag regression analysis and composite analysis. The lead-lag regression identifies how much time

lag exists between the perturbation of a climate mode and the response of the UTLMS water vapor anomalies at different pressure levels. We normalized each index by dividing each index anomaly by its standard deviation before performing the linear regressions. For every pressure level and time shift, a univariate linear regression is performed first with respect to either ENSO or QBO index individually (WV = $X_0$ + $X_1 \times$ENSO and WV = $X_0$ + $X_1 \times$QBO). The respective

R-squared value of each linear regression, a standard measure of proportion of explained variance, is used to indicate how much water vapor variability can be described by each linear regression separately for each pressure level and time lag. The maximum R-squared value will determine the optimal time lag for the univariate linear regression at each pressure level. A multivariate linear regression with respect to ENSO and QBO together is then performed using

the optimal time lags obtained from the univariate linear regressions to estimate how much water vapor variability can be described by ENSO and QBO combined. The residual between the original observation and the multivariate linear regression with respect to ENSO and QBO together is also calculated to quantify how much water vapor variability that cannot be explained by ENSO and QBO together and may be due to nonlinear or coupled ENSO-QBO

interaction and other physical processes (e.g., BDC). We recognize that using the R-squared value of a linear regression as proportion of explained variance is based on the following strong

assumption: the UTLMS water vapor interannual anomaly is a linear function of ENSO or QBO index with a Gaussian distribution. Within the observed interannual anomalies of the UTLMS water vapor and climate variabilities, these assumptions are not perfect but useful.

For the composite analysis, we first partitioned the interannual monthly MLS water vapor anomalies into four different cases based on different combinations of ENSO and QBO phases: warm ENSO (MEI > 0.3) and westerly QBO (u50 > 0.1 m s$^{-1}$) case, warm ENSO (MEI > 0.3) and easterly QBO (u50 < -0.1 m s$^{-1}$) case, cold ENSO (MEI < -0.3) and westerly QBO (u50 > 0.1 m s$^{-1}$) case, and cold ENSO (MEI < -0.3) and easterly QBO (u50 < -0.1 m s$^{-1}$) case. These

threshold values were chosen in order to remove the ENSO and QBO neutral phases and have sufficient samples for the composites at the same time. We then averaged the interannual monthly MLS water vapor anomalies for each case to create a composite mean profile. The composite analysis was applied to the MLS interannual water vapor anomaly data for annual means, the summer months (MJJASO) average and the winter months (NDJFMA) average

separately.

## 3 Results

Figure 2 shows the interannual monthly mean tropical water vapor anomalies from MLS in percentage deviations at different pressure levels from 215 hPa to 6 hPa and from August 2004 to September 2017. In the upper troposphere from ~215 hPa to ~120 hPa, large vertically

oriented tropical water vapor anomalies of ±15% are evident. They seem to be coincident with several El Nino or La Nina events shown in Fig. 1 with positive anomalies during the warm ENSO phases and negative anomalies during the cold ENSO phases. In the lower and middle stratosphere (100-6 hPa), large tropical water vapor anomalies of ±15% are found to propagate upward at a speed of about 7 km per year starting around 100 hPa with a first local maximum

in the lower stratosphere around 68 hPa and a second local maximum in the middle stratosphere around 15 hPa. These have been referred to as the interannual variability of the stratospheric water vapor tape recorder (e.g., Geller et al., 2002; Kawatani et al., 2014; Liang et al., 2011) and are regulated by QBO. The small interannual water vapor anomalies at the beginning and ending months of the data record are results of the boundary effect of using the difference of running means as a band pass filter.

To show the relative importance of ENSO and QBO and their roles in the interannual tropical water vapor anomalies at different pressure levels, Figure 3 shows the R-squared values for the linear regressions between the MLS tropical UTLMS interannual monthly water vapor anomalies and the ENSO or QBO index from 215 hPa to 6 hPa with time lag shifts from 0-24 months. Figure 3a (left) is for ENSO ($WV = X_0 + X_1 \times ENSO$) and Fig. 3b (right) is for QBO ($WV = X_0 + X_1 \times QBO$). The time lag shift indicates the number of months that the tropical water vapor anomalies lag the ENSO or QBO index. The maximum time lag of 24 months was chosen due to the fact that 24 months are close to the period of a QBO cycle and the minimum period of an ENSO cycle. Figure 3a indicates the R-squared value for the linear regressions between the tropical water vapor anomalies and the ENSO index is large (~0.5) in the upper troposphere, becomes smaller (~0.1) at the tropopause (~100 hPa) and is very small (close to zero) in the stratosphere above 80 hPa. This is consistent with the large vertically oriented water vapor anomalies of ±15% in the upper troposphere that are coincident with several El Nino or La Nina events shown in Fig. 2. This implies that ENSO has a strong impact on the water vapor interannual variability in the upper troposphere and around the tropopause, while its impact on the water vapor interannual variability is very small in the stratosphere. The current finding of the strong impact of ENSO on the water vapor in the upper troposphere and around the tropopause is consistent with several previous studies (Dessler et al., 2014; Gettelman et al., 2001; Liang et al., 2011; Ye et al., 2018) that suggested ENSO can strongly

modulate the upper tropospheric water vapor through the convective transport of tropospheric water vapor and the evaporation of cloud ice. In terms of the response time, the highest correlation is at ~3-month lag. This is comparable to the tropospheric temperature response time to the ENSO SST anomaly (Su et al., 2005). The current finding of the weak influence of ENSO on the water vapor anomalies in the lower and middle stratosphere is also consistent with a previous study (Ding and Fu, 2018) that suggested the small effect of ENSO on tropical zonal mean lower stratospheric water vapor is due to the opposite phases of lower stratospheric water vapor anomalies in response to ENSO in the longitudinal direction. This is due to the compensating effect of ENSO on the tropical tropopause temperature anomalies between the western equatorial Pacific and the central equatorial Pacific that reduces the zonal mean tropical tropopause temperature anomalies found by previous studies (Avery et al., 2017; Gettelman et al., 2001; Kiladis et al., 2001; Liang et al., 2011).

For the linear regressions between the interannual tropical water vapor anomalies and the QBO index, Figure 3b indicates that the QBO influence is small in the upper troposphere but large in the lower and middle stratosphere. The high R-squared value between the tropical water vapor anomalies and the 50-hPa QBO index starts at the tropopause at a time lag of ~2 months and propagates upwards, peaking in the middle stratosphere at ~15 hPa with a time lag of ~21 months, and disappearing at about 6 hPa. The R-squared value is large (~0.5) at the tropopause at a time lag of ~2 months. Above 100 hPa, it first decreases to about 0.3 at about 40 hPa at a time lag of ~13 months. Above 40 hPa, it then increases to about 0.5 in the middle stratosphere at about 15 hPa with a time lag of ~21 months (a local peak). Above 15 hPa, it then decreases again till it disappears at about 6 hPa. The peak at ~15 hPa with a time lag of ~8 months is the result of the upward propagating signal starting at the tropopause at a time lag of a few months earlier than the 50-hPa QBO index. These phase lags are consistent with the findings of previous studies (Dessler et al., 2013; 2014; Ye et al., 2018). The time lag of ~2 months for the

high correlation between the tropical tropopause water vapor anomalies and the 50-hPa QBO index is similar to the time needed for the QBO signal to propagate downward from the 50-hPa level to the tropopause (Tweedy et al., 2017). The above vertical structure of the high R-squared value for the linear regressions between the tropical water vapor anomalies and the 50-hPa

QBO index suggests that the QBO does not directly affect the water vapor concentration at altitudes higher than 100 hPa, instead the QBO signal in the tropical LMS water vapor is imprinted at the tropopause (about 100 hPa) first and then it is transported upward with the tape recorder. This observational evidence supports that the QBO's impact on the stratospheric water vapor is from its modulation on the tropical tropopause temperature, as suggested by

many previous studies (e.g., Diallo et al., 2018; Fueglistaler et al., 2009; Liang et al., 2011; Randel et al., 2004; Tweedy et al., 2017; Zhou et al., 2001, 2004). This is also consistent with the upward propagating tropical water vapor anomalies of $\pm 15\%$ in the lower and middle stratosphere shown in Fig. 2 and referred to as the interannual variability of the stratospheric water vapor tape recorder (e.g., Geller et al., 2002; Liang et al., 2011). This indicates that the

interannual variability of the stratospheric water vapor tape recorder (e.g., Geller et al., 2002; Liang et al., 2011) is a result of the impact of QBO. It is possible that the interannual variations of the BDC play a role here too because the QBO modulates the extratropical wave activity, an important driver for the BDC, which influences the tropical cold point tropopause temperature.

Between about 120 hPa and about 40 hPa, the decrease of the high R-squared value with altitude can be easily understood by mixing or dilution of the upward transport of the imprinted 100-hPa signal by the tape recorder. However, the increase of the high R-squared value between 40 hPa and 15 hPa and the peak of the high R-squared value at about 15 hPa cannot easily be explained by the tape recorder only. Other sources, such as the downward propagation

of water vapor anomalies in the upper stratosphere due to the methane oxidation, may be responsible for this phenomenon (Kawatani et al., 2014).

Figure 4 shows the time series of the monthly interannual tropical MLS water vapor anomalies (blue lines) and the predicted interannual monthly tropical water vapor anomalies based on the univariate linear regressions on the ENSO index only (red lines) or the QBO index only (orange lines) at the time lag less than 12 months with the highest R-squared value for four specific pressure levels: 147-hPa, 100-hPa, 68-hPa, 15-hPa, representing the upper troposphere, tropopause, lower stratosphere, and mid-stratosphere, respectively. Figure 4 reaffirms the results shown in Fig. 3: the decreasing contributions of ENSO and the increasing contributions of QBO to the interannual variability of UTLMS water vapor as the altitude increases. The R-squared value for the linear regressions between the MLS interannual tropical water vapor anomalies and the ENSO index is ~0.54 at the 147-hPa altitude with a ~3-month time lag, ~0.08 at the tropopause with a ~11-month time lag, and deceases to ~0.01 at the 68-hPa altitude with a ~12-month time lag and ~0.02 at the 15-hPa altitude with a ~6-month time lag. In contrast, the R-squared value for the linear regressions between the MLS interannual tropical water vapor anomalies and the QBO index is small (~0.03) at the 147-hPa altitude with a ~0-month time lag, becomes significant and large (~0.46) at the tropopause with a ~2-month time lag, (~0.44) at the 68-hPa altitude with a ~5-month time lag and (~0.52) at the 15-hPa altitude with a ~8-month time lag. Therefore, for the interannual variability of the UTLMS tropical water vapor, ENSO is more important than QBO in the upper troposphere (from ~215 hPa to ~120 hPa), both ENSO and QBO are important around the tropopause (from ~110 hPa to ~90 hPa), and mainly by QBO is more important than ENSO in the lower and middle stratosphere (from ~80 hPa to 6 hPa). This result is consistent with many previous studies. For example, Tweedy et al. (2017) and Diallo et al. (2018) have shown that the sudden shift in the QBO from westerly to easterly wind shear in the boreal winter of 2015–2016 significantly decreased global lower stratospheric water vapor from early spring to late autumn and reversed the lower stratosphere

moistening to the lower stratosphere drying. Their results imply that QBO is more important than ENSO in modulating the lower-stratospheric water vapor.

Figure 4 also shows the predicted monthly interannual tropical water vapor anomalies based on the multivariate linear regressions on the ENSO and QBO indices together (purple lines) and the differences (green lines) between the original MLS interannual monthly tropical water vapor anomalies (blue lines) and the multivariate linear regression (purple lines) at the four specific pressure levels. Fig. 4 indicates that the predicted interannual monthly tropical water vapor anomalies based on the multivariate linear regressions of ENSO and QBO are very similar to the original MLS interannual monthly tropical water vapor anomalies. ENSO and QBO together can explain about more than half (~50-60%) variance of the interannual monthly tropical water vapor anomalies under the strong assumption that the UTLMS water vapor interannual anomaly is a linear function of ENSO or QBO index with a Gaussian distribution. However, large residues are still evident in Fig. 4 indicating that nonlinear ENSO-QBO interactions and other physical processes (e.g., BDC) as well as their interactions may be considered in order to explain the full interannual variability of the tropical UTLMS water vapor. This issue is beyond the scope of this paper and will be investigated in the future.

To highlight the different roles of ENSO and QBO phases in the interannual tropical water vapor anomalies at different pressure levels and different seasons, Figure 5 shows the composite interannual tropical water vapor anomalies from MLS as function of pressure levels for winter (NDJFMA) (blue lines), summer (MJJASO) (red lines), and annual (black lines) means at four different cases based on different combinations of ENSO and QBO phases. Consistent with Figs. 3 and 4, Figure 5 shows that ENSO mainly impacts the interannual tropical water vapor anomalies in the upper troposphere and at the tropopause, while QBO mainly affects the interannual tropical water vapor anomalies at the tropopause and in the lower

and middle stratosphere. In the upper troposphere (215-120 hPa), the interannual tropical water vapor anomalies are mainly related by the ENSO phase and its seasonal change while the QBO's effect seems to be small. Positive interannual tropical water vapor anomalies are found during the warm ENSO phases, while negative interannual tropical water vapor anomalies are found during the cold ENSO phases for the annual, winter and summer means no matter what QBO phases are. However, exception exist for the summer and the cold easterly QBO case but the sampling for this case is low and we have to interpret this result with caution. This is consistent with the aforementioned mechanism that ENSO impacts the upper tropospheric water vapor through the convective transport of water vapor (e.g., Jiang et al., 2015). The interannual tropical water vapor anomalies tend to be larger during the winter than during the summer because the ENSO events are usually stronger during the winter than during the summer (Wallace et al., 1998).

Near the tropopause (110-90 hPa), both ENSO and QBO as well as their season changes can influence the interannual tropical water vapor anomalies. Both warm ENSO phase and westerly QBO phase tend to cause positive interannual tropical water vapor anomalies while cold ENSO phase and easterly QBO phase tend to cause negative interannual tropical water vapor anomalies in this layer. As a result, different interannual tropical water vapor anomalies are found for different cases depending the different ENSO and QBO phase combinations and their seasonal variations in this layer (Liang et al., 2011). For example, very strong positive interannual tropical water vapor anomalies are found for the warm westerly case and very strong negative interannual tropical water vapor anomalies are found for the cold easterly case due to the supporting effect of ENSO and QBO for the winter season. Weak interannual tropical water vapor anomalies are found for the warm easterly case and the cold westerly case due to the compensating effect of ENSO and QBO. This result is consistent with those results found by Diallo et al. (2018) and Liang et al. (2011) that emphasized the importance of the interaction

of ENSO and QBO phases in controlling the tropical tropopause water vapor anomalies. However, Diallo et al. (2018) focused on the lower stratosphere, different from the tropopause layer we discussed here.

In the lower and middle stratosphere (80-6 hPa), QBO and its seasonal change contributes significantly to the interannual tropical water vapor anomalies while the ENSO's effect is negligible. QBO explains ~50-60%, in contrast to ~2% by ENSO, of the tropical water vapor interannual variance under the strong assumption that the UTLMS water vapor interannual anomaly is a linear function of ENSO or QBO index with a Gaussian distribution. As discussed earlier, this result seems to be consistent with Tweedy et al. (2017) and Diallo et al. (2018) but not with Avery et al. (2017). During the westerly QBO phases, interannual tropical water vapor anomalies are positive near the tropopause and in the lower stratosphere (below ~50-hPa altitude), negative in the lower and middle stratosphere (between ~50-hPa and ~20-hPa altitude), and positive again in the middle stratosphere (above ~20-hPa altitude) for all seasons. The opposite occurs during the easterly QBO phases. The sign reversal of the interannual tropical water vapor anomalies along the pressure levels in Fig. 5 are consistent with the upward propagating water vapor anomalies and the R-squared values shown in Figs. 2 and 3b. There are some differences in interannual tropical water vapor anomalies between warm and cold ENSO phases and between the summer and winter seasons, but they are relatively small.

**4 Summary and Conclusions**

In this study, we have analyzed the Aura MLS tropical UTLMS monthly water vapor data from 215 hPa to 6 hPa and from August 2004 to September 2017 using time-lag regression analysis and composite analysis to explore the interannual variations of water vapor in the whole

tropical UTLMS layer and their connections to ENSO and QBO. The main findings of our analysis are summarized below.

In the upper troposphere (215-120 hPa), ENSO and its seasonal change contributes significantly to the interannual tropical water vapor anomalies with a ~3-month time lag while the QBO's effect is negligible. ENSO explains ~54%, in contrast to ~3% by QBO, of the interannual tropical water vapor variance under the strong assumption that the UTLMS water vapor interannual anomaly is a linear function of ENSO or QBO index with a Gaussian distribution. ENSO modulates the upper tropospheric water vapor mainly through the convective transport of tropospheric water vapor and the evaporation of cloud ice. Positive interannual tropical water vapor anomalies are found during the warm ENSO phases, while negative interannual tropical water vapor anomalies are found during the cold ENSO phases for all seasons although the interannual tropical water vapor anomalies tend to larger during the winter than during the summer.

Near the tropopause (110-90 hPa), both ENSO and QBO as well as their seasonal changes are important for the interannual tropical water vapor anomalies. ENSO explains ~8% while QBO explains ~46% of the interannual tropical water vapor variance under the strong assumption that the UTLMS water vapor interannual anomaly is a linear function of ENSO or QBO index with a Gaussian distribution. ENSO can modulate the tropical tropopause water vapor through the convective transport of tropospheric water vapor, the evaporation of cloud ice, and its impact on the tropical tropopause temperature. In contrast, QBO modulates the tropical tropopause water vapor mainly by its modulation of the tropical tropopause temperature. Both warm ENSO phase and westerly QBO phase tend to cause positive interannual tropical water vapor anomalies while both cold ENSO phase and easterly QBO phase tend to cause negative interannual tropical water vapor anomalies. As a result, different interannual tropical water

vapor anomalies are found for different combinations of ENSO and QBO phases and their seasonal variations. For example, very strong positive interannual tropical water vapor anomalies are found for the warm westerly case and very strong negative interannual tropical water vapor anomalies are found for the cold easterly case due to the compatible effects of ENSO and QBO for the winter season. Weak interannual tropical water vapor anomalies are found for the warm easterly case and the cold westerly case due to the compensating effects of ENSO and QBO. This emphasized the importance of the interaction of ENSO and QBO phases in controlling the tropical tropopause water vapor anomalies.

In the lower and middle stratosphere (80-6 hPa), QBO and its seasonal change contributes significantly to the interannual monthly water vapor anomalies while the ENSO's effect is negligible. QBO explains ~50-60%, in contrast to ~2% by ENSO, of the interannual tropical water vapor variance under the strong assumption that the UTLMS water vapor interannual anomaly is a linear function of ENSO or QBO index with a Gaussian distribution. QBO modulates the tropical lower and middle stratospheric water vapor mainly by its modulation of the tropical tropopause temperature. During the westerly QBO phase, interannual tropical water vapor anomalies are positive near the tropopause and in the lower stratosphere (below ~50-hPa altitude), negative in the lower and middle stratosphere (between ~50-hPa and ~20-hPa altitude), and positive again in the middle stratosphere (above ~20-hPa altitude) for all seasons. The opposite occurs during the easterly QBO phase. There are some small differences in interannual tropical water vapor anomalies between warm and cold ENSO phases and between the summer and winter seasons.

In summary, ENSO has a strong impact on the interannual variations of tropical water vapor below 90-hPa altitude, i.e., in the upper troposphere and at the tropopause. On the other hand, QBO has a large impact on the interannual variations of tropical water vapor above 110-hPa,

i.e., at the tropopause and in the lower and middle stratosphere. ENSO and QBO together can explain more than half (~50-60%) but not all the interannual variations of the tropical UTLMS water vapor. Nonlinear ENSO-QBO interactions and other physical processes (e.g., BDC) as well as their interactions may be considered in future investigations in order to fully explain the interannual variability of the tropical UTLMS water vapor.

The findings in the current study are generally consistent with those from previous studies (e.g., Dessler et al., 2014; Diallo et al., 2018; Ding and Fu, 2018; Liang et al., 2011; Tweedy et al., 2017; Ye et al., 2018). However, the relative roles of ENSO and QBO on the tropical UTLMS water vapor interannual variabilities for the entire UTLMS layer and at different phase lags and different pressure levels are more completely investigated in the current study than the previous ones. In addition, this study provides direct empirical evidence to support a belief that the QBO impacts the tropical UTLMS water vapor mainly through its influence on the tropical tropopause temperature. These results can serve as an important benchmark for future climate model evaluation studies.

**Data availability**

The MLS water vapor data used in this research are freely available through the Aura MLS project website (https://mls.jpl.nasa.gov). The multivariate ENSO indices used in this research are freely available from the NOAA Earth System Research Laboratory (ESRL) website (https://www.esrl.noaa.gov/psd/enso/mei/). The 50-hPa QBO indices used in this research are also freely available from the NOAA NCEP Climate Prediction Center (CPC) website (http://www.cpc.ncep.noaa.gov/data/indices/qbo.u50.index).

## Author contributions

JHJ, HS, and BT designed this study. EWT performed the data analysis and prepared the figures. All authors contributed to the discussion of the results and the preparation of the manuscript.

## Competing interests

The authors declare that they have no conflict of interests.

## Acknowledgments

This research was performed at Jet Propulsion Laboratory, California Institute of Technology (Caltech), under a contract with National Aeronautics and Space Administration. It was supported by the Aura Microwave Limb Sounder project. The authors thank Dr. Andrew
Dessler, Dr. Michaela Hegglin, and two anonymous referees for their constructive and insightful comments that helped to improve the quality of this paper. The first author also thanks Matthew Worden for his help with the MATLAB code. Caltech Copyright 2019. All rights reserved.

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

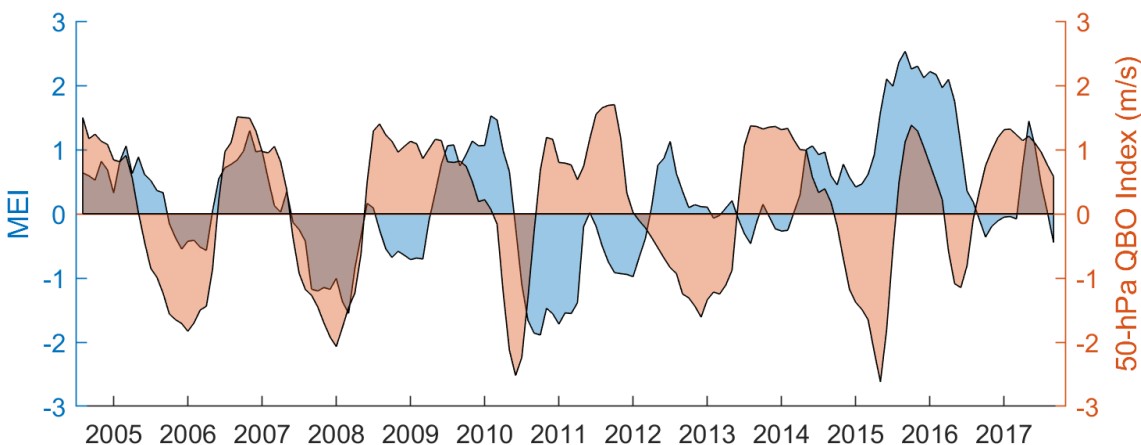

**Figure 1.** Bimonthly multivariate ENSO index (MEI, blue) and monthly 50-hPa QBO index (u50, m s$^{-1}$, orange) based on standardized anomaly of monthly zonal mean zonal wind at the Equator and 50-hPa both from NOAA at the period from August 2004 to September 2017. Positive MEI values indicate warm ENSO (El Nino) phases while negative MEI values indicate cold ENSO (La Nina) phases. Positive u50 values denote westerly QBO phases while negative u50 values denote easterly QBO phases.

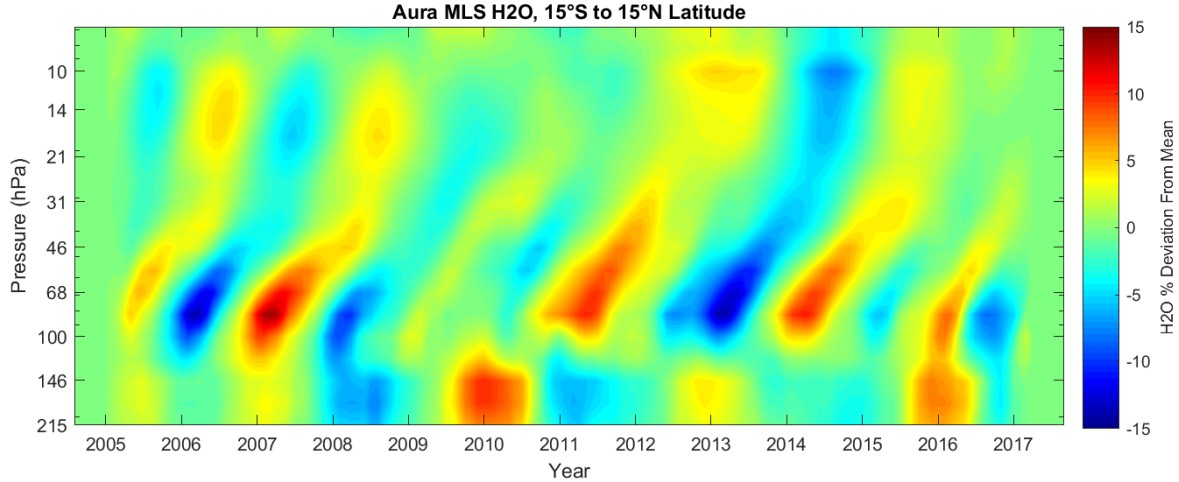

**Figure 2.** Monthly interannual tropical water vapor anomalies from MLS in percentage deviations at different pressure levels from August 2004 to September 2017.

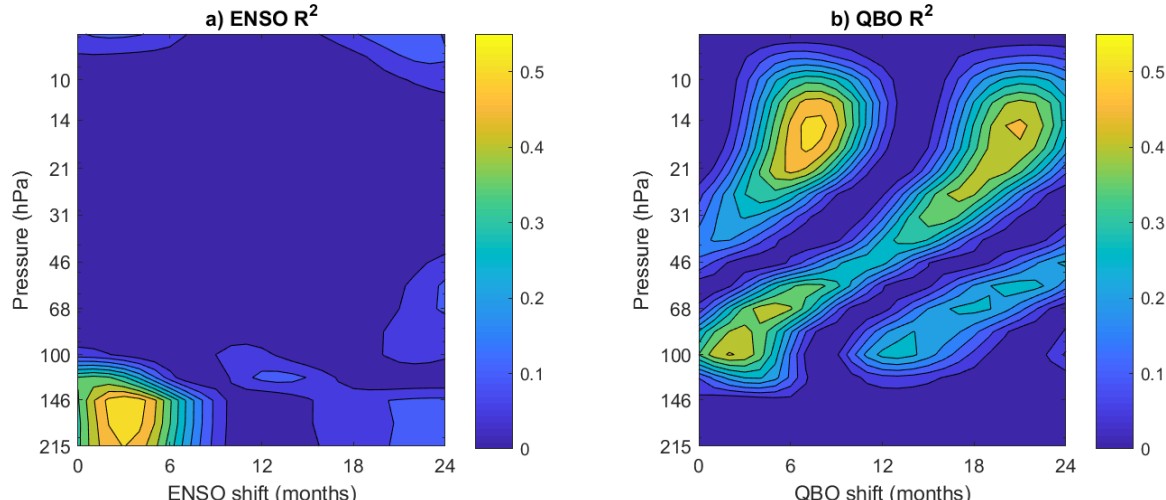

**Figure 3.** R-squared values for the linear regressions between the interannual tropical water vapor anomalies from MLS and the ENSO or QBO index at each pressure level with time lag shifts from 0-24 months. Figure 3a (left) is for ENSO (WV = $X_0$ + $X_1 \times$ENSO) and Fig. 3b (right) is for QBO (WV = $X_0$ + $X_1 \times$QBO). The time lag shift indicates the number of months that interannual tropical water vapor anomalies lag the ENSO or QBO index.

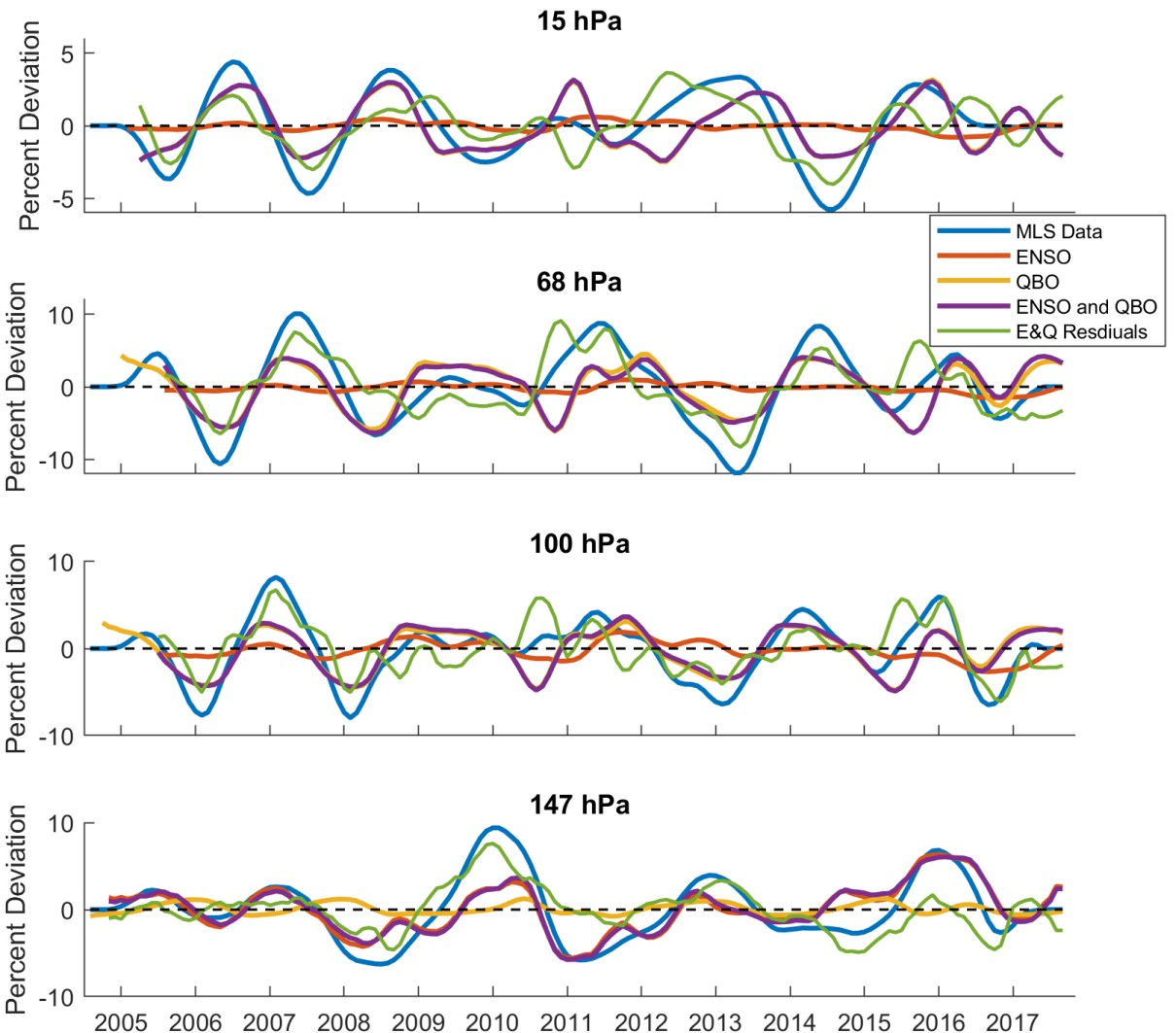

**Figure 4.** The time series of the monthly interannual tropical water vapor anomalies from MLS (blue lines) and the predicted monthly interannual tropical water vapor anomalies based on the linear regressions on the ENSO index only (red lines), the QBO index only (orange lines), and the ENSO and QBO indices together (purple lines) at the time lag less than 12 months with the highest R-squared value for four specific pressure levels: 15 hPa (top row), 68 hPa (second row), 100 hPa (third row), and 147 hPa (bottom row). The differences between the MLS data (blue lines) and the linear regression lines based on the ENSO and QBO together (purple lines) are also plotted (green lines).

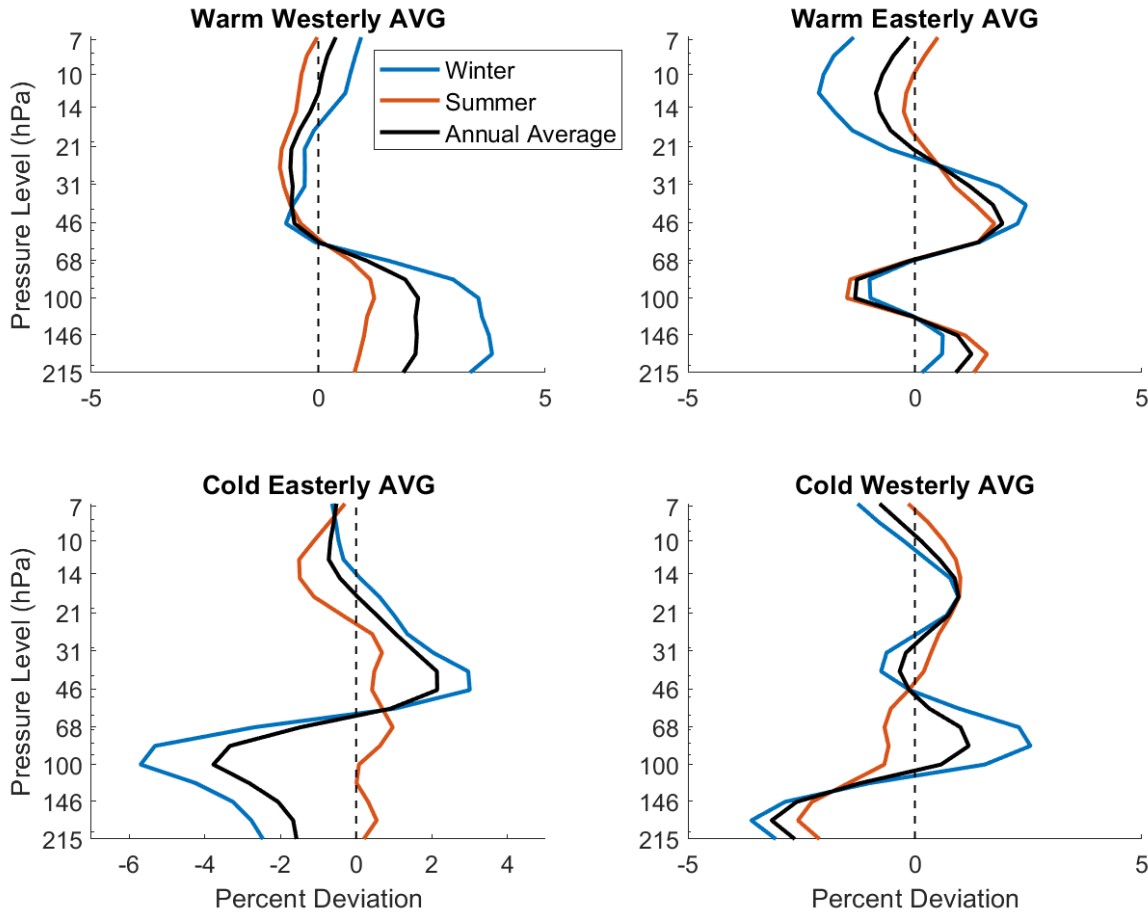

**Figure 5.** Composite interannual tropical water vapor anomalies from MLS at different pressure levels for winter (NDJFMA) (blue lines), summer (MJJASO) (red lines), and annual (black lines) means at four different cases based on different combinations of ENSO and QBO phases.