# Peer review of "Interannual variations of water vapor in the tropical upper troposphere and the lower and middle stratosphere and their connections to ENSO and QBO"

_Atmospheric Chemistry and Physics, 2018_

## Short Comment (SC1) · 5 Dec 2018

A paper by Diallo et al. (2018) recently published in ACP discusses the combined influence of QBO and ENSO on the UTLS water vapour and ozone distributions using a lagged multiple regression analysis. A reference to this paper should be included along with a discussion of the added scientific value of the results presented here.

Diallo, M. et al. (2018) Response of stratospheric water vapor and ozone to the unusual timing of El Niño and the QBO disruption in 2015–2016. Atmospheric Chemistry and

Physics, 18 (17). pp. 13055-13073. ISSN 1680-7316

also accessible via

https://www.atmos-chem-phys.net/18/13055/2018/

---

## Referee Comment (RC1) · Anonymous Referee #2 · 11 Dec 2018

The manuscript by E.W. Tian et al is a very focused study on an interesting, however, very narrow, topic which is in the scope of ACP. The paper is, on the whole, well organized and written in a very clear style. Some major issues I raised in the context of the initial submission have been successfully remedied before this discussion paper has been resubmitted. My major concern is that the predictors analyzed (ENSO and QBO) explain only a fraction of the observed H2O anomalies and, in more general terms, that only incremental new evidence is provided for relations between ENSO and QBO, which, as far as I can judge, have been known before. To make the paper acceptable

for ACP, it will be necessary to highlight which new insights have been gained.

More specifically: On p4 l22-24 it is stated that "some fundamental physical or dynamical processes controlling UTLMS water vapor and its variability are not well represented or even missing in the climate models and reanalyses". However, since only about half of the $H_2O$ anomalies is explained by QBO and ENSO (p12 l3-4), the results do not provide any clue what the key to solving the problem with the models and reanalyses actually is.

p. 5 l. 19: Are these uncertainties 1 sigma or two sigma? Are these uncertainties used in the manuscript? Do these uncertainties survive the averaging process mentioned on p. 6 l4 because they are chiefly systematic or are they random and thus cancel largely out during the averaging?

p. 6 l 10 -15: The method how short-handed anomalies are isolated through the difference between 12-month and 42-month running means is not clear to me.

On p. 7 l 13 a normalization of the indices is mentioned but the rationale behind this action is not clear. Isn't normalization implicit part of each correlation analysis?

On p. 7 l 14 linear regression analysis is mentioned. Have the authors investigated if the relatively poor explanatory and predictive power of the regression model used might be due to the assumed linearity? Could it be that ENSO or QBO have nonlinear influence on water vapor? Only from the fact that linear correlation with ENSO and QBO indices do not satisfactorily explain the observation it cannot be concluded that other processes are needed. Nonlinear interaction or coupled ENSO-QBO interaction has not been ruled out.

p9 l1-2: The explanation of small interannual water vapor anomalies at the first and last several months of the data record by limitations of the band pass filter is not very clear. Does this refer to the running means mentioned before? Is this simply a boundary effect occuring where the width of the filter exceeds the range where data are available? Isn't

a running mean a low-pass filter rather than a band-pass filter?

p. 12 l 3-4 and elsewhere: The relation between R**2 and the explained variance ho;ds only under certain assumptions (linearity, Gaussian distributions). A critical discussion of this issue is needed.

---

## Referee Comment (RC2) · Anonymous Referee #1 · 4 Feb 2019

**1 Summary**

This paper reports on the impact of ENSO and QBO on tropical UTLS and MS water vapor (WV) variability in the tropics using a longer observational data record from MLS. The paper can be followed easily, however, there are some serious issues and some clarifications needed. Most of the results (Figs 1-4) are not very new and basically confirming previous studies, only Fig. 5 showing the composite anomalies for the different combination of QBO (eeasterly/westerly) and ENSO phases (warm/cold) are

nice results but not unexpected.

**2 Major issues**

- When talking of time lag between QBO and WV, this lag is here defined with the respect to the 30 hPa QBO index. This may differ for QBO indices at other altitude levels. A quantitative statement is needed how the lag changes when changing the pressure level of the QBO index. The comparison with previous studies are hampered as the latter apparently mostly used the 50 hPa index. So it would be recommendable to use the 50hPa index. The authors did not provide a justification why the 30hPa QBO index was their preferred choice.

- A weakness of the paper is that the the linear regression is mainly done with respect to QBO and ENSO only. This is reasonable for obtaining the optimized time lag. The $r^2$ values barely reach 0.5 for each of the indices alone, so it would be interesting to have a more complete multiple regression attempting to explain more completely the WV variances (as compared to the MLR limited to QBO and ENSO). It is evident from Fig. 4 that the residuals (after removing the QBO and ENSO components) still show large variances. A more complete MLR with additional factors may enhance the detectibility of the QBO and ENSO factors, when they are rather weak. In the introduction the effect of the BDC is mentioned as one important factor for WV variability. Even though BDC is related to QBO, the Dressler et al. 2014 paper gets a high correlation in their regression model combining BDC, QBO, and ENSO (or Delta T) with observations at least for 82 hPa.

[Figure]

**3 Minor issues**

p. 1, l. 20: "The phase lag is based on the 30-hPa QBO index and should be different from that found by previous studies based on the 50-hPa QBO index" The results presented in the paper regarding the phase lag should be reported in a way that can be directly compared with previous studies, so I strongly suggest to use the 50-hPa QBO index instead (see major comments).

p. 2., l. 12: Another good reference on the strong water vapor feedback is Riese et al. (doi:10.1029/2012JD017751, 2012, see their Fig. 1)

p. 3., l. 19: "... impact on the tropical UTLMS water vapor is mainly through the QBO's influence on the tropical tropopause temperature that regulates the amount of upper tropospheric water vapor entering the stratosphere". Here it would be good to mention that the BDC plys a role here as well since the QBO modulates the extratropical wave activity, an important driver for the BDC, that influences the tropical cold point tropopause temperature.

p. 4., l. 4: "They found that the evaporation of convective ice from increased deep convection as the troposphere warms plays an important role in the tropopause water vapor variability". The sentence before suggests that this was derived from MLR applied to observations, but indeed Ye et al. (2014) used a combination of observations and models to come to this conclusion.

p. 6, l. 21: The MEI ENSO index is a two-month average, whence it is a lagged index itself. This should be mentioned here.

p. 8, l. 14: "monthly mean tropical water vapor anomalies" (add "mean")

p. 8, l. 25: "... may be regulated by QBO" Is there any doubt that QBO is one of the main driver of this variability, so I suggest to use something else than "may be".

p. 10, l. 16: change "suggests" to "confirms" (as this is in agreement with earlier

studies).

p. 10, l. 24: " as result of the impact from the QBO". One should mention here that BDC also plays a role here (see my earlier comment).

p. 11, l. 18: change "by mainly QBO instead of ENSO" to "mainly by the QBO"

p. 11, l. 23: discussion of the green line in Figure 4. As discussed in the major points, the residuals show large variability, so an improved MLR could minimize this variance and improve the signal form the QBO and ENSO, where they are weak.

p. 12, l. 7: "This issue is beyond the scope of this paper and will be investigated in the future." I disagree here as this should be done here to improve the paper (see major comments).

p. 12, l. 22: "interannual tropical water vapor anomalies tend to be larger" (add "be")

Fig. 1: units for the wind is missing, "standard departure" should be replaced by "standard deviation".

Fig. 3.: physical units (months) for shifts are missing, color legend has no label ($r^2$).

---

## Author Comment (AC1) · 17 Feb 2019

We appreciate Dr. Hegglin and two anonymous referees for their constructive and insightful comments. We have revised and improved our paper based on these comments listed in *blue italic font* below. Our detailed responses to these comments are described and listed in black regular font below. The page numbers are based on the revised version in the tracked change form.

Response to Dr. Hegglin

*A paper by Diallo et al. (2018) recently published in ACP discusses the combined influence of QBO and ENSO on the UTLS water vapour and ozone distributions using a lagged multiple regression analysis. A reference to this paper should be included along with a discussion of the added scientific value of the results presented here.*
*Diallo, M. et al. (2018) Response of stratospheric water vapor and ozone to the unusual timing of El Niño and the QBO disruption in 2015–2016. Atmospheric Chemistry and Physics, 18 (17). pp. 13055-13073.*

We have referenced this relevant study and discussed the added scientific value of our results relative to this and other references. Please see the added text on pages 7, 19 and 22 in the revised version.

Response to Anonymous Referee #1

*1 Summary*

*This paper reports on the impact of ENSO and QBO on tropical UTLS and MS water vapor (WV) variability in the tropics using a longer observational data record from MLS. The paper can be followed easily, however, there are some serious issues and some clarifications needed. Most of the results (Figs 1-4) are not very new and basically confirming previous studies, only Fig. 5 showing the composite anomalies for the different combination of QBO (easterly/westerly) and ENSO phases (warm/cold) are nice results but not unexpected.*

We have resolved and clarified the serious issues the referee raised. Please see our response to the major issues below.

*2 Major issues*

*• When talking of time lag between QBO and WV, this lag is here defined with the respect to the 30 hPa QBO index. This may differ for QBO indices at other altitude levels. A quantitative statement is needed how the lag changes when changing the pressure level of the QBO index. The comparison with previous studies are hampered as the latter apparently mostly used the 50 hPa index. So it would be recommendable to use the 50hPa index. The authors did not provide a justification why the 30hPa QBO index was their preferred choice.*

We have performed additional analyses, replotted Figs. 1, 3, 4 and 5 and revised our paper accordingly using the 50-hPa QBO index downloaded from the NOAA CPC website. The changes can be found through the paper. For example, we have added the following discussions on page 1: "The phase lag is based on the 50-hPa QBO index used by many previous studies."

We have added the following discussions on page 11:

"For QBO, we use the standardized anomaly of monthly zonal mean zonal wind at the Equator and 50-hPa (u50, m s$^{-1}$) based on the National Centers for Environmental Prediction (NCEP) / National Center for Atmospheric Research (NCAR) reanalysis (CDAS) downloaded from NOAA NCEP Climate Prediction Center (CPC) website (http://www.cpc.ncep.noaa.gov/data/indices/qbo.u50.index). Positive u50 values denote westerly QBO phases while negative u50 values denote easterly QBO phases. This 50-hPa QBO index has been frequently used by previous studies (Dessler et al., 2013; 2014; Ye et al., 2018)."

*• A weakness of the paper is that the the linear regression is mainly done with respect to QBO and ENSO only. This is reasonable for obtaining the optimized time lag. The r2 values barely reach 0.5 for each of the indices alone, so it would be interesting to have a more complete multiple regression attempting to explain more completely the WV variances (as compared to the MLR limited to QBO and ENSO). It is evident from Fig. 4 that the residuals (after removing the QBO and ENSO components) still show large variances. A more complete MLR with additional factors may enhance the detectibility of the QBO and ENSO factors, when they are rather weak. In the introduction the effect of the BDC is mentioned as one important factor for WV variability. Even though BDC is related to QBO, the Dressler et al. 2014 paper gets a high correlation in their regression model combining BDC, QBO, and ENSO (or Delta T) with observations at least for 82 hPa.*

We agree that the paper would be enhanced if a more complete Multiple Linear Regression (MLR) with additional factors, such as the Brewer-Dobson Circulation (BDC), as suggested by this referee. However, we feel this is out of the scope of this study for the following reasons. First, the main purpose of this paper to highlight the relative roles of ENSO and QBO in the interannual variations of water vapor in the tropical UTLMS layer at different levels. Second, there are many factors that impact the interannual variations of the tropical UTLMS water vapor, such as ENSO, QBO, BDC, Hadley circulation, aerosols, and methane. Thus, it is difficult to perform a complete MLR with all contributing factors. For example, Dessler et al. (2014) has shown that a more complete MLR including ENSO, QBO and BDC can get a higher correlation but only increase the R-squared value to about 0.7-0.8, still less than perfect. Third, the ENSO and QBO indices have been well constrained observationally and used extensively. In contrast, the BDC index is usually inferred indirectly from the behavior of material tracers and is plagued by large uncertainties (Butchart, 2014). Thus, the regression based on the BDC may contain inherent errors. Due to the above reasons, we prefer a linear regression based on ENSO and QBO only in this paper and a more complete MLR with additional factors can be deferred to future studies.

*3 Minor issues*

*p. 1, l. 20: "The phase lag is based on the 30-hPa QBO index and should be different from that found by previous studies based on the 50-hPa QBO index" The results presented in the paper regarding the phase lag should be reported in a way that can be directly compared with previous studies, so I strongly suggest to use the 50-hPa QBO index instead (see major comments).*

We have resolved this issue. Please see our response to major issue above.

*p. 2., l. 12: Another good reference on the strong water vapor feedback is Riese et al. (doi:10.1029/2012JD017751, 2012, see their Fig. 1)*

We have added this reference. Please see changes on page 2 in the revised version.

*p. 3., l. 19: "... impact on the tropical UTLMS water vapor is mainly through the QBO's influence on the tropical tropopause temperature that regulates the amount of upper tropospheric water vapor entering the stratosphere". Here it would be good to mention that the BDC plys a role here as well since the QBO modulates the extratropical wave activity, an important driver for the BDC, that influences the tropical cold point tropopause temperature.*

We have added the following discussions "It is also noted that the QBO modulates the extratropical wave activity, an important driver for the BDC, which influences the tropical cold point tropopause temperature." Please see changes on page 5 in the revised version.
We have also added some discussions on BDC on pages 3 and 4.

*p. 4., l. 4: "They found that the evaporation of convective ice from increased deep convection as the troposphere warms plays an important role in the tropopause water vapor variability". The sentence before suggests that this was derived from MLR applied to observations, but indeed Ye et al. (2014) used a combination of observations and models to come to this conclusion.*

We have added 'based on satellite observations and model simulations' in the sentence before the sentence "They found...". Please see changes on page 6 in the revised version.

*p. 6, l. 21: The MEI ENSO index is a two-month average, whence it is a lagged index itself. This should be mentioned here.*

We have added 'The MEI is computed separately for each of twelve sliding bi-monthly seasons (Dec/Jan, Jan/Feb, ..., Nov/Dec).  We use the MEI value of month(i-1) and month(i) as if it were the value for month(i) only as advised by the MEI author.' Please see changes on page 11 in the revised version.

*p. 8, l. 14: "monthly mean tropical water vapor anomalies" (add "mean")*

We have added 'mean'. Please see changes on page 13 in the revised version.

*p. 8, l. 25: "... may be regulated by QBO" Is there any doubt that QBO is one of the main driver of this variability, so I suggest to use something else than "may be".*

We have changed 'may be' to 'are'. Please see changes on page 13 in the revised version.

*p. 10, l. 16: change "suggests" to "confirms" (as this is in agreement with earlier studies).*

We have changed 'suggests to 'confirms'. Please see changes on page 15 in the revised version.

*p. 10, l. 24: " as result of the impact from the QBO". One should mention here that BDC also plays a role here (see my earlier comment).*

We have added 'It is possible that the interannual variations of the BDC play a role here too because the QBO modulates the extratropical wave activity, an important driver for the BDC, which influences the tropical cold point tropopause temperature." Please see changes on page 15 in the revised version.

*p. 11, l. 18: change "by mainly QBO instead of ENSO" to "mainly by the QBO"*

Corrected. Please see changes on page 16 in the revised version.

*p. 11, l. 23: discussion of the green line in Figure 4. As discussed in the major points, the residuals show large variability, so an improved MLR could minimize this variance and improve the signal form the QBO and ENSO, where they are weak.*

Please see our response to major issue above.

*p. 12, l. 7: "This issue is beyond the scope of this paper and will be investigated in the future." I disagree here as this should be done here to improve the paper (see major comments).*

Please see our response to major issue above.

*p. 12, l. 22: "interannual tropical water vapor anomalies tend to be larger" (add "be")*

Corrected. Please see changes on page 18 in the revised version.

*Fig. 1: units for the wind is missing, "standard departure" should be replaced by "standard deviation".*

we have replotted Fig.1 using new data of the 50-hPa QBO index and added 'MEI' and '50-hPa QBO Index (m/s)' in the y-axis titles. Please see changes on Fig.1 in the revised version.

*Fig. 3.: physical units (months) for shifts are missing, color legend has no label (r2).*

We have replotted Fig.3 using new data of the 50-hPa QBO index and based on this suggestion. Please see changes on Fig.3 in the revised version.

Response to Anonymous Referee #2

*The manuscript by E.W. Tian et al is a very focused study on an interesting, however, very narrow, topic which is in the scope of ACP. The paper is, on the whole, well organized and written in a very clear style. Some major issues I raised in the context of the initial submission have been successfully remedied before this discussion paper has been resubmitted. My major concern is that the predictors analyzed (ENSO and QBO) explain only a fraction of the observed H2O anomalies and, in more general terms, that only incremental new evidence is provided for relations between ENSO and QBO, which, as far as I can judge, have been known before. To make the paper acceptable for ACP, it will be necessary to highlight which new insights have been gained.*

We have highlighted new insights gained from the current study in the introduction and conclusion. For example, we have added the following discussions on page 8: "This study distinguishes itself from previous similar studies in three following ways: (1) The current study investigates the interannual variations of water vapor in the whole tropical UTLMS layer from 215 hPa to 6 hPa instead of a couple of layers in the previous ones. (2) The Aura MLS UTLMS water vapor data of much longer length (August 2004 to September 2017) are used in the current study than the previous ones. (3) The relative importance of ENSO and QBO on the tropical UTLMS water vapor interannual variabilities for the entire UTLMS layer and at different phase lags are more completely investigated in the current study than the previous ones."

*More specifically: On p4 l22-24 it is stated that "some fundamental physical or dynamical processes controlling UTLMS water vapor and its variability are not well represented or even missing in the climate models and reanalyses". However, since only about half of the H2O anomalies is explained by QBO and ENSO (p12 l3-4), the results do not provide any clue what the key to solving the problem with the models and reanalyses actually is.*

We have deleted this paragraph. Please see changes on page 8 in the revised version.

*p. 5 l. 19: Are these uncertainties 1 sigma or two sigma? Are these uncertainties used in the manuscript? Do these uncertainties survive the averaging process mentioned on p. 6 l4 because they are chiefly systematic or are they random and thus cancel largely out during the averaging?*

We have added the following discussion on page 9 in the revised version:
"These measurement uncertainties are retrieval uncertainties and estimated based on (1) the average difference between the simulated retrieval and truth file; (2) the average difference between MLS measurements and the air borne measurements. These uncertainties should not affect our results because we are interested in the interannual anomalies instead of its means. In addition, Hegglin et al. (2013) show that MLS zonal monthly mean water vapor show very

good to excellent agreement with the multi-instrument mean (MIM) in comparison between 13 instruments, throughout most of the atmosphere (including the UTLS) with mean deviations from the MIM between +2.5% and +5%, making these random errors irrelevant for the average monthly zonal mean water vapor anomalies used in this study (Diallo et al., 2018)."

*p. 6 l 10 -15: The method how short-handed anomalies are isolated through the difference between 12-month and 42-month running means is not clear to me.*

We have deleted a period that was misplaced in this sentence. Hope this sentence is clearer now to readers. Using the difference of running means of different widths as a band-pass filter is effective. A 12-month running mean will remove the high-frequency variabilities (<2 years) and keep the interannual variability (2-7 years) and the low-frequency variabilities (>7 years). A 42-month running mean will remove the high-frequency variabilities (<2 years) and the interannual variability (2-7 years) and keep the low-frequency variabilities (>7 years). As a result, the difference between the 12-month and 42-month running means will remove both the high-frequency (<2 years) variabilities and the low-frequency (>7 years) variabilities and keep the interannual variability (2-7 years) only. This simple approach of band pass filter has been used in the previous studies related to the Madden-Julian Oscillation (e.g., Tian et al., 2006; 2007; 2011). Please see the changes on page 10 in the revised version.

*On p. 7 l 13 a normalization of the indices is mentioned but the rationale behind this action is not clear. Isn't normalization implicit part of each correlation analysis?*

We agree that normalization is an implicit part of a correlation analysis and we do not need to perform the index normalization first if for a correlation analysis only or a univariate linear regression only. However, a multivariate linear regression with respect to ENSO and QBO together is also performed in this study. Thus, the normalized ENSO and QBO indices are needed. The normalization is also used in previous studies, such as Diallo et al. (2018).

*On p. 7 l 14 linear regression analysis is mentioned. Have the authors investigated if the relatively poor explanatory and predictive power of the regression model used might be due to the assumed linearity? Could it be that ENSO or QBO have nonlinear influence on water vapor? Only from the fact that linear correlation with ENSO and QBO indices do not satisfactorily explain the observation it cannot be concluded that other processes are needed. Nonlinear interaction or coupled ENSO-QBO interaction has not been ruled out.*

We agree that nonlinear or coupled ENSO-QBO interaction could be responsible for the unexplained variance of water vapor. However, we have not investigated on this issue yet. It is also possible other processes (e.g., BDC) and the non-linear interactions among ENSO, QBO and BDC could be responsible for the unexplained variance of water vapor. This issue is beyond the scope of this paper and will be investigated in the future. We stated this possibility in the paper on pages 12, 17 and 22 in the revised version.

*p9 l1-2: The explanation of small interannual water vapor anomalies at the first and last several months of the data record by limitations of the band pass filter is not very clear. Does this refer to the running means mentioned before? Is this simply a boundary effect occurring where the width of the filter exceeds the range where data are available? Isn't a running mean a low-pass filter rather than a band-pass filter?*

Yes, the band pass filter refers to the difference of running means mentioned on Section 2. This is simply a boundary effect occurring where the width of running means exceeds the range where data are available. A running mean is a low-pass filter rather a band-pass filter. However, as we explained earlier the difference of running means of different widths can act as a simple band-pass filter. Because of the boundary effect the running means will have no effect in the beginning and ending parts of time series. As a result, the difference of the running means will be very small. We have stated the following on page 13 in the revised version: "The small interannual water vapor anomalies at the beginning and ending months of the data record are results of the boundary effect and limitation of using the difference of running means as a band pass filter."

*p. 12 l 3-4 and elsewhere: The relation between R\*\*2 and the explained variance holds only under certain assumptions (linearity, Gaussian distributions). A critical discussion of this issue is needed.*

The referee is right that the relation between R\*\*2 and the explained variance is based on linearity and Gaussian distributions. Although we don't expect ENSO or QBO influences on the UTLMS water vapor are strictly linear, it is useful to assume the responses of water vapor are approximately proportional to the magnitudes of ENSO or QBO anomalies within the observable ranges of perturbations. It is also reasonable to assume the interannual anomalies of water vapor and ENSO or QBO have Gaussian distributions. We realized that these assumptions are not perfect but useful. We have added a couple of sentences to state the underlying assumptions on page 12 in the revised version.

References

[revised manuscript text omitted]

---

## Author Response (AR2)

Dear Dr. Stiller,

Thank you very much for providing us another opportunity to further address the reviewes' concerns and revise the maniuscript. We appreciate the comments and suggestions from you and the anonymous referee #2 during this round of review. We have further revised and improved our paper based on these comments listed in *blue italic font* below. Our detailed responses to these comments are described in black regular font below. The page numbers are based on the revised version in the tracked change form. We hope that you will find this version acceptable for publication.

Response to Anonymous Referee #2

*This is a review of the revised manuscript. The two most critical issues of the original version of the manuscript were the lack of new results and the weak predictive power of the linear regression presented. I think that these problems have not been solved; thus I cannot recommend publication of this manuscript in its present form. The material presented, however, seems to contain more information than exploited by the authors and I think that another major revision could make the paper acceptable.*

We appreciate your comments and suggestions and we have further revised and improved our paper based on your comments and suggestions. We hope that you will find this version acceptable for publication although we agree that the results presented here are only incremental instead of fundamentally new.

*1. The weak explanatory and predictive power of the regression analysis:*

*In the abstract (p1, l20) the authors claim that interannual tropical water vapor anomalies are MAINLY determined by ENSO... is clearly false. This wording suggests that a major fraction of the variability can be explained, but the R\*\*2 values suggest (assuming Gaussian statistics) that not even half of the variation is explained. At the very least a more modest wording is needed throughout the paper in a sense that ENSO and QBO do make contributions (of varying weight) but that more than half of the variation remains unexplained. The authors refuse to strive for a more complete explanation of the water vapor variabilities and they offer arguments to justify this decision. I agree with the authors insofar as a multilinear regression is indeed not needed if the goal is not to explain the water vapor variability. The fact that only a fraction of the variability is explained needs to be clearly stated in the paper. The question remains what, if not the explanation of the water vapor variability, shall be main the scientific content of the paper. This leads me to the second major point:*

Thank you for your detailed comments. We agree that a more modest wording is needed through the paper and we have revised the paper accordingly. We have stated clearly in the paper that only a fraction of the variability is explained by ENSO and QBO. For example, in the abstract and conclusion, we stated the following: "Our analysis shows that the interannual tropical UTLMS water vapor anomalies are strongly related to ENSO and QBO which together can explain more than half (~50-60%) but not all variance of the interannual tropical water vapor anomalies."

We also agree and want to strive for a more complete explanation of the water vapor variabilities as well. However, one major obstacle for us to perform such analyses is that the first author who did all the analyses last summer is back to his school and busy with his school courses now.

*2. The lack of new results:*

*The authors claim that the goal of the paper is to determine the relative weight of QBO and ENSO as a driver for water vapor concentrations. This, as such is not new, and the authors argue that they use a longer time series and a better altitude coverage than preceding work. With this, the added value is only incremental.*

We agree our results are only incremental in nature instead of fundemntally new but are still worth a publication for updated results with newer and longer datasets than used in existing publications.

*The authors analyze the time lag between the QBO proxy as a function of altitude but fall short of interpreting this result in a sense of providing a scientific explanation of this relation. This relation, however, may be the key to new insights into the driving mechanisms and may help to deviate from worn paths.*

Thank you for pointing this out. We have highlighted this new insight in the revised paper, such as the introduction (page 8), the text (pages 15-16) and the conclusion (page 23).

*My proposed chain of arguments runs as follows:*

*1. The slope of the time lag with altitude (Fig 3) matches that of the water vapor anomaly (Fig 2) pretty well. This suggests that the QBO does NOT directly affect the water vapor concentration at altitudes higher than 100 hPa but that the QBO signal is imprinted at about 100 hPa and is transported upward with the tape recorder. This does not come as a surprise from a theoretical point of view but I am not sure if any empirical evidence has yet presented for this. It is clear that the tape recorder transports any imprinted signal upwards, but a specific evidence that this holds for the QBO part of the variation may be new. If not such things, what else would be the benefit that the analysis covers more altitudes than previous studies?*

Thank you for your detailed comments. We have highlighted this new insight in the revised paper, such as the introduction (page 8), the text (pages 15-16) and the conclusion (page 23).

*2. Between about 120 hPa and about 40 hPa R\*\*2 seems to decrease with altitude. This is plausible and can be understood by mixing, dilution, a leaky pipe or whatever.*

*3. Interestingly, R\*\*2 has a maximum at about 14-20 hPa. This cannot easily be explained by transport of an imprinted 100 hPa signal by the tape recorder. This phenomenon begs for an explanation and deserves a thorough discussion of related literature in order to identify possible explanations.*

Thank you for your detailed comments. We have add some discussion of the difference between the level from 120 hPa to 40 hPa and the level above 40 hPa in the text (page 16).

*I think the other studies still can remain in the paper. There is nothing wrong with them but they seem to me to be quite a routine application of research tools without leading anywhere. In combination with the suggestion above, they still provide useful additional incremental knowledge.*

Thank you for your suggestion to keep other results in this paper. We agree our results are only incremental in nature instead of fundemntally new but are still worth a publication just like many other pulications are.

*3. Specific comments:*

*p2 l15 shouldn't the references be listed in chronological order?*

Based on the ACP manuscript preparation guidelines for authors ([https://www.atmospheric-chemistry-and-physics.net/for_authors/manuscript_preparation.html](https://www.atmospheric-chemistry-and-physics.net/for_authors/manuscript_preparation.html)), the order of in-text citations can be based on relevance, as well as chronological or alphabetical listing, depending on the author's preference. Thus, we think the current alphabetical order is fine for ACP.

*p9 l13/14 but what if the biases included in the uncertainty estimates are time-, temperature- or whatever dependent?*

This is possible but no evidence suggests this is the case.

*p12 l15-18 is there any evidence that these assumptions are justified? Histograms of similar? If the residual variance is not a random effect but caused by unknown forcing, the assumption of a Gaussian distribution may not be adequate.*

We agree on this point but we do not have evidence for this assumption yet.

*p21 l5 I find it audacious to claim that ENSO and QBO explain most of the variation, facing the fact that R\*\*2 hardly exceed 0.5. "Make significant contributions" would be more adequate.*

We agree and we have changed the text accordingly.

*Fig 1: Axes captions too small*

We have replotted Fig. 1 with larger axes captions.

*Fig 4/5: Legend too small*

We have replotted Figs. 4&5 with bigger legends.

[revised manuscript text omitted]

---

## Author Response (AR3)

Dear Dr. Stiller,

We appreciate the anonymous referee #2 for his/her further comments listed in *blue italic font* below. We have further revised our paper and made the necessary corrections based on these comments. Our detailed responses to these comments are described in black regular font below. The page numbers are based on the revised version in the tracked change form. Thank you very much for accepting this paper for publication in ACP.

Response to Anonymous Referee #2

*The most recent revised version of the paper by Tian et al. has considerably improved. I still would have preferred a more comprehensive paper because the data seem to contain more information than explored in this paper but I think the current content of the paper is sufficient for publication in ACP after some minor corrections listed below.*

Thank you very much for your suggestions and efforts.

*1. It seems unconventional to me to report R\*\*2 values as percentage values. They should be reported as pure numbers between 0 and 1. In the context of "explained variance", however, percentage values are fine. The determination coefficient (R\*\*2) and the explained variance are not the same, thus it seems inadequate to me to report the determination coefficient in a way in which usually the explained variance is reported. Just under certain conditions there happens to be a one to one relationship such that explained var. = R\*\*2 \* 100 %.*

We have reported the R-squared values as pure numbers between 0 and 1 as you suggested. Please see the changes in the revised paper on pages 14, 15, and 17.

*2. Whenever the explained variance is inferred from the R\*\*2 values, the underlying assumptions (chiefly linearity) should be stated.*

We have stated the underlying assumption when we discuss the proportion of explained variance using the R-squared value. Please see changes in the revised paper on pages 13, 18, 20, 21, and 22.

[revised manuscript text omitted]